# Reclaimed Rubber/Poly(ε-caprolactone) Blends: Structure, Mechanical, and Thermal Properties

**DOI:** 10.3390/polym12051204

**Published:** 2020-05-25

**Authors:** Aleksander Hejna, Łukasz Zedler, Marta Przybysz-Romatowska, Javier Cañavate, Xavier Colom, Krzysztof Formela

**Affiliations:** 1Department of Polymer Technology, Faculty of Chemistry, Gdańsk University of Technology, Gabriela Narutowicza 11/12, 80–233 Gdańsk, Poland; aleksander.hejna@pg.edu.pl (A.H.); lukzedle@student.pg.edu.pl (Ł.Z.); marta.przybysz@pg.edu.pl (M.P.-R.); 2Department of Chemical Engineering, Universitat Politècnica de Catalunya Barcelona Tech, Carrer de Colom, 1, 08222 Terrassa, Barcelona, Spain; francisco.javier.canavate@upc.edu

**Keywords:** ground tire rubber, reclaimed rubber, poly(ε-caprolactone), waste management, modification, melt-compounding

## Abstract

The amount of elastomeric waste, especially from tires is constantly increasing on a global scale. The recycling of these residua should be considered a priority. Compounding the waste rubbers with other polymers can be an excellent alternative to reuse waste materials. This procedure requires solving the issue of the lack of compatibility between the waste rubber particles and other polymers. Simultaneously, there is a claim for introducing biodegradable plastics materials to reduce their environmental impact. In this work, reclaimed rubber/poly(ε-caprolactone) (RR/PCL) blends are proposed to enhance the recycling and upcycling possibilities of waste rubbers. The results show that the addition of PCL to the RR allows obtaining blends with improved mechanical properties, good thermal stability, and enhanced interfacial compatibility between the used components. Structure and properties of the proposed RR/PCL have been studied by means of static and dynamic mechanical testing, Fourier transform infrared spectroscopy (FTIR), differential scanning calorimetry (DSC), and thermogravimetric analysis (TGA)-FTIR analysis.

## 1. Introduction

According to the published data, the amount of elastomeric waste, especially from tires, grows annually in the world [1,2,3,4]. The recycling of these residua, produced after providing a useful service life of the original materials, is considered as a long-term solution, superior to the other end-of-life options, as landfill or incineration. In the pursuit of possible new uses, some methods and technologies to enhance recycling and upcycling, have been used.

The main recycling route for waste tires consists of converting the tires into ground tire rubber (GTR) by a process of shredding, separation of components, and granulation [5]. This product is nowadays available on the market and is usually the main source of the recycling initiatives for that material. Some studies explore the possibility of application of GTR in the processes that can generate scission of the chains aiming to break the cross-linked structure of the rubber. Some of these processes are mechanical, thermal or chemical and generally, are not selective, affecting the disulfide bonds created by the vulcanization but also the macromolecular chains. The GTR treated by this method is called reclaimed rubber (RR).

Blending the elastomeric waste with other polymers can be an excellent alternative to reuse the waste materials, as long as there is good compatibility between the components. Since achieving that compatibility is the key point, several processes have been used to improve the compatibility in blends [6,7,8]. Most of these processes include pretreatments, additives or modifications of the ground tire rubber in order to develop compatibility with the matrixes. The morphology of the GTR particles is also very significant to improve the compatibility with the matrix. Colom et al. [9] demonstrated that etching GTR with oxidizing acids improves the ability of the rubber to interact with the high-density polyethylene (HDPE), finding that the treatment with H_2_SO_4_ was the most effective, whereas HClO_4_ did not improve the material’s properties.

The possibilities of blending elastomeric waste have been extended to other types of materials. In this sense, an interesting approach was proposed by Formela et al. [10]. Using waste from used tires, ground tire rubber (GTR) and low-density polyethylene (LDPE), they created LDPE/GTR blends including also a non-polar elastomer. This polymeric blend was then used as a modifier to bitumen. The conducted investigations concluded that the properties of the bitumen could be improved through this modification based on the addition of GTR. Similar observations were reported by Yan et al. [11], who studied the effect of the addition of GTR, and recycled LDPE on rheological properties of asphalts. Bitumen not only has been used as a matrix to include the different waste modifiers. Zedler et al. [12] studied the mechano-chemical modification of GTR in blends with road bitumen and assessed remarkable synergistic effects in the devulcanization of GTR by microwaves.

Another interesting approach seems to be the application of waste rubbers in biodegradable or partially biodegradable polymeric materials. This solution should reduce the negative impact of waste rubbers, especially end-of-life tires on the environment and human health. However, the published data about research in this area are rather limited [13,14]

Poly(ε-caprolactone) (PCL) is a linear aliphatic polyester polymer with interesting properties. Some of its features are semi-crystallinity, biodegradability, non-toxicity, and good mechanical properties. PCL has been used to prepare films [15] and composite materials for several applications [16,17]. These applications include zinc oxide, which is a well-known and important semi-conductor used in sensors, solar cells, optoelectronics, and other devices. Likewise, ZnO is an important component in the cross-linking system used during vulcanization of rubber compounds. The possibility of integrating a biodegradable component that at the same time can interact with the ZnO or zinc stearate present in GTR can constitute a novel approach to the thermoplastic blends. Moreover, this phenomenon can explain why the addition of ZnO to PCL/GTR blends results in improved dielectric constant and dielectric loss capacities in comparison to blends with CuO or CuO_0.5_–ZnO nanoparticles, what was recently described by Varaprasad et al., who also describes the existence of interactions between PCL and ZnO based on Fourier transform infrared spectroscopy (FTIR) and thermogravimetric analysis (TGA) results [18]. Other interesting applications of blends including GTR and PCL, are related to the possibility of improvement of the processing of the rubber waste because the addition of PCL, which makes these materials interesting for filtration membranes [19,20] or low-cost flexible electronic materials [21,22].

As commented above, the mechano-chemical modification of GTR with bitumen also presents interesting possibilities to improve the interaction between components. All these considerations led to the idea of novel blends presented in this study.

In this paper, we proposed new blends based on a reclaimed rubber (RR) made of ground tire rubber modified with bitumen, which was subsequently blended with a biodegradable poly(ε-caprolactone). To better understand the RR and PCL interfacial interactions, two types of PCL with different molecular weights were investigated. The microstructural changes and their influence in the compatibility between components have been assessed by FTIR and thermogravimetric analysis combined with FTIR. Moreover, the mechanical properties of RR/PCL blends, such as toughness, tensile strength, elongation at break, have been analyzed to determine their potential applications. Blends between biodegradable polymers and ground tire rubber may constitute a suitable approach to the circular economy of the waste rubbers.

## 2. Experimental

### 2.1. Materials

Ground tire rubber (GTR) obtained by ambient grinding of used tires (a combination of passenger car and truck tires in 50:50 mass ratio) with particles size below 0.8 mm, produced by Orzeł S.A. (Poniatowa, Poland) was used during research.

Bitumen modified with styrene–butadiene block copolymer—Modbit 25/55–60 with penetration at 25 °C: 25–55 (1/10 mm) and softening point: ≥60 °C was received from Lotos Asfalt Sp. z o.o. (Gdansk, Poland).

Two types of poly(ε-caprolactone)—Capa™ 6800 and Capa™ FB100—were provided by Perstorp (Malmö, Sweden). The first one is a linear polyester with a molecular weight of 80,000 g/mol and a melting point in the range of 58–60 °C. The second is a partly cross-linked polyester with a molecular weight of 100,000 g/mol and a melting point in the range of 58–60 °C. The properties of the applied PCL types are presented in Table 1.

### 2.2. Sample Preparation

#### 2.2.1. Mechano-Chemical Reclaiming of GTR

Ground tire rubber (GTR) was mechano-chemically treated in the presence of bitumen in the ratio of 100:10. Reclaiming was performed on two-roll mills (diameter: 200 mm and working space: 400 mm) at ambient temperature. The following two-roll mills settings were applied: friction equaled 1.08 and the gap width varied between 0.2 and 3 mm. The time of treatment was 15 min and the obtained reclaimed rubber (RR) showed a continuous and homogenous structure. Sol fraction of reclaimed rubber was 15 wt.%. More detailed information about the impact of bitumen on low-temperature reclaiming was published elsewhere [12,23]. The main components of ground tire rubber are natural rubber, synthetic rubber (presence of two peaks confirmed by differential thermogravimetric analysis), curing additives and carbon black. Total content of polymers and organic additives is ~60–65 wt.%, while content of carbon black and inorganic additives is ~30–35 wt.% estimated by thermogravimetric analysis. Moreover, it should pointed out that precise composition of post-consumer waste as GTR (e.g., ratio between natural rubber to synthetic rubber) due to complex composition of waste tires and their mixture prior grinding is unknown.

#### 2.2.2. Reclaimed Rubber/Poly(ε-caprolactone) Blends (RR/PCL) Preparation

Reclaimed rubber/poly(ε-caprolactone) blends were prepared in a Brabender mixer (Brabender type GMF 106/2, Duisburg, Germany) at 120 °C, where screw rotation was set to 80 rpm. The mixing time was 8 min. During the first 2 min, PCL was plasticized and then the reclaimed rubber (RR) was added. The mixing with reclaimed rubber was continued for the next 6 min. Obtained materials were molded into 2-mm thick samples under the pressure of 4.9 MPa at 120 °C for 1 min and then at room temperature for 5 min. For modification of RR two types of PCL were applied in the amount of 10 to 50 wt.%. Unmodified samples comprising solely of RR were not prepared, because its preparation according to the above-mentioned procedure was impossible. The application of PCL enabled the processing of the material at 120 °C.

### 2.3. Measurements Preparation

The density of studied blends was measured based on the Archimedes method according to ISO 1183. All measurements were performed in methanol at room temperature.

The chemical structure of the prepared materials was determined using Fourier transform infrared spectroscopy (FTIR) analysis performed using a Nicolet Spectrometer IR200 from Thermo Scientific (Waltham, MA, USA). The device had attenuated total reflectance (ATR) attachment with a diamond crystal. Measurements were made with 1 cm^−1^ resolution in the range from 4000 to 400 cm^−1^.

The melt flow index of the materials was determined using Zwick flow plastometer (Zwick Roell Group, Ulm, Germany) according to ISO 1133 at 230 °C, with a load of 10 kg.

The tensile strength and elongation at break were obtained in accordance with ASTM D638. Tensile tests were performed on the Zwick Z020 apparatus (Zwick Roell Group, Ulm, Germany) at a constant speed of 50 mm/min. Shore hardness type D was measured using Zwick 3131 durometer (Zwick Roell Group, Ulm, Germany) in accordance with ISO 868.

The dynamic mechanical analysis was performed using the DMA Q800 TA Instruments apparatus (New Castle, DE, USA). Samples cut to the dimensions of 40 × 10 × 2 mm were loaded with a variable sinusoidal deformation force in the single cantilever bending mode at the frequency of 1 Hz under the temperature rising rate of 4 °C/min within the temperature range between −100 and 100 °C.

Results of the static and dynamic mechanical analysis were used to calculate the brittleness of investigated materials, in accordance with the following formula presented by Brostow et al. [24] (1):(1)$$B=\frac{1}{\varepsilon_{b}\times E^{\prime }}$$
where: *B*–brittleness, 10^10^ %·Pa; ε*_b_*–elongation at break, %; *E’*–storage modulus at 25 °C, MPa.

The thermal analysis was performed using TGA by means of the model Q600 from TA Instruments (New Castle, DE, USA). Samples of blends weighing approx. 10 mg were placed in a corundum dish. The study was conducted in an inert gas atmosphere - nitrogen (flow rate 100 mL/min) in the range from 25 to 800 °C with a temperature increase rate of 20 °C/min. Volatile products from thermal degradation were also evaluated using a Fourier transform infrared spectroscopy (FTIR). During TGA measurements, volatile degradation products were directed (using a heated transfer line at 220 °C) to a Nicolet iS10 spectrometer from Thermo Scientific (Waltham, MA, USA). This solution allows the “on-line” characterization of the volatile products during the TGA measurements. The timing offset of FTIR spectra comparing to TGA curves is related to the volume of the thermogravimetric apparatus chamber.

The thermal behavior and crystallization of the samples were measured by differential scanning calorimetry (DSC) measurement was carried out on a DSC 204 F1 Phoenix apparatus (Netzsch Group, Selb, Germany). The samples of 8–9 mg weight were placed in an aluminum pan and heated from 20 to 100 °C under nitrogen atmosphere at a rate of 10 °C/min, subsequently material was cooled from 100 to −80 °C at a rate of 10 °C/min and heated to 100 °C at a rate of 10 °C/min. The results from second heating were discussed.

The DSC thermograms with the melting points (T_m_), crystallization temperature (T_c_), enthalpy of crystallization and enthalpy of melting were recorded during the second heating. For all samples, the degree of crystallization of PCL phase (X_cPCL_) was calculated according to the following Equation (2):(2)$$X_{cPCL}=\left(\frac{\Delta H_{m}}{\Delta H_{0}\times W_{f}}\right)\times 100\%$$
where Δ*H*_m_ is the specific melting enthalpy, Δ*H*_0_ is the melting enthalpy of 100% crystalline virgin polymer (where the melting enthalpy of 100% PCL is 136 J/g [25]), and *W*_f_ is the weight fraction of PCL in RR/PCL blends.

There are also presented values of the supercooling parameter, calculated according to the following formula (3) [26]:(3)$$\Delta T_{\alpha }=T_{m}-T_{c}$$
where: *T_m_*—melting temperature, °C and *T_c_*—crystallization temperature, °C.

The increase in the supercooling parameter for studied samples, together with the decrease in crystallization and melting temperatures, points to the reduction of nucleating activity of PCL phase in RR/PCL blends.

The morphology of the tensile fracture surfaces created by breaking the samples in the tensile test at the speed of 500 mm/min was investigated by using a HITACHI model S3400 (Tokyo, Japan) scanning electron microscope (SEM).

## 3. Results and Discussion

### 3.1. Physical Properties

Table 2 shows the values of the density of RR/PCL blends. The influence of the PCL type on this parameter was rather low, which was predictable given their similar densities, 1.132 for Capa™ 6800 and 1.130 g/cm^3^ for Capa™ FB100. The density at 210 °C, calculated from the results of melt flow analysis, also shows similar results. Additionally, the values of the porosity of the samples are presented. They were determined from the experimental and theoretical density, calculated using the rule of a mixture based on the Equation (4):(4)$$\rho =\rho_{RR}\left(1-X\right)+\rho_{PCL}X$$
where: *r*—density, g/cm^3^, *X*—a fraction of the PCL, while subscripts RR and PCL are related to the reclaimed rubber and applied PCL.

Porosity has a significant influence on the mechanical properties of polymer materials. Even when the differences between particular samples are not very large, the porosity may affect significantly the mechanical performance. Wang et al. [27] demonstrated that an increase of porosity from 5.84% to 6.32% in poly(lactic acid) results in a decrease in Young’s modulus from 2970 to 2800 MPa.

The porosity of the samples has a range from 1.82% to 2.65% in the samples prepared with Capa™ 6800 and from 1.47% to 2.38% when using Capa™ FB100. Nevertheless, the influence of the type of PCL (with similar density—see Table 1) on the porosity of obtained materials was very small. The slight differences in porosity between samples obtained with the two types of PCL can be explained by complex composition of reclaimed rubber based on waste tires.

### 3.2. Rheological Properties

The structure of the PCL macromolecules had a more significant impact on the rheological properties of samples. In Figure 1, the results of the melt flow analysis are presented. The values show that the influence of the PCL on the processing of the RR is quite evident. The poor flowability of the samples containing 10 wt.% of PCL made impossible to perform the MFI measurements. The flowability for MFI tests, became acceptable only for RR/PCL blends with 20 wt.% or higher content of PCL. Following this way, the amount of PCL in the samples increases substantially the melt flow rate which goes from limited flow (RR/PCL samples in ratio 90/10) at all up to 9–14 cm^3^/10min (RR/PCL samples in ratio 50/50).

As said above, the type of PCL had a noticeable impact on the RR flow rate. Capa™ 6800 reports more flow ability to the samples than Capa™ FB100. This is related to respective differences in MFI value, 32.16 cm^3^/10 min for Capa™ 6800 vs. 18.34 cm^3^/10 min for Capa™ FB100. These differences are related to the molecular weight and morphology of polymer chains, which stimulates flow disturbances and reduces melt volume flow rate (MVR) and melt mass flow rate (MFR) parameters [28]. According to the differences in their structure, the impact of PCL type becomes more significant at higher contents.

### 3.3. Static Mechanical Properties

Table 3 shows the values of static mechanical properties determined by means the tensile and hardness tests. Both types of poly(ε-caprolactone) resulted in an enhancement of the mechanical performance of RR. An increase of PCL content from 10 to 50 wt.% led to an increase of tensile strength of 190% in the case of Capa™ 6800 and 212%, for Capa™ FB100. The increase of tensile strength can be directly related to the very good tensile strength of PCL, which is 25.1 MPa for Capa™ 6800 and 42.5 MPa for Capa™ FB100, respectively, and the compatibility between the reclaimed rubber and PCL phases. The improvement of the other mechanical properties could be also related to the content in PCL and good interaction between both components.

Moreover, the values of the permanent set (see Table 3), allow observing a difference between both types of PCL. The values of the permanent set are higher for samples containing Capa™ FB100. As has been mentioned previously, this behavior is related to differences in the structure of both PCL types and the breaking of the partially cross-linked structure of Capa™ FB100 during tensile tests. For low contents of PCL, the differences between both types of studied blends are rather slight, because PCL acts mainly as a plasticizer of RR particles, making the excellent interfacial interactions between PCL and RR, and the mechanism of stress transfer inside the material. At higher contents, PCL becomes a continuous phase and the mechanical properties of blends increase as a function of the amount of modifier. In these terms, Capa™ FB100, with its partially cross-linked structure, provides higher mechanical values than Capa™ 6800.

The stress–strain curves obtained during the tensile testing of the materials were also used to calculate the toughness of the material, which measures the total amount of energy that can be absorbed by the RR/PCL blends before failure. Figure 2 shows schematically the integration of the stress–strain curve resulting in a quantitative determination of that energy. To exhibit high values of toughness, the material must be able to withstand high stress and elongation. Therefore, toughness can be understood as a combination of strength and high ductility. In the presented RR/PCL blends, the best combination of these features was detected when using Capa™ FB100.

Following the general trend of the other studied properties, the modification of RR with both types of poly(ε-caprolactone) defines a significant increase of hardness, which, as in the previous discussion, was ascribed to the properties of applied PCL. For 50 wt.% content of PCL, obtained hardness was 92.6 and 92.9 ShA, respectively for Capa™ 6800 and Capa™ FB100. Neat PCL showed values of 95.9 and 96.9 ShA. This means that despite the 50 wt.% share of RR, hardness was at the levels of 96.6 and 95.9% of PCL hardness. This phenomenon confirms the observations made during tensile tests, meaning that at higher contents of PCL, the modifiers starts acting as a continuous phase and determines the mechanical performance of the new RR/PCL blends.

The same reasoning could be applied to all the parameters determined during the static mechanical tests. At lower PCL contents, PCL acts mainly as a modifier for RR particles. Therefore, the increase of mechanical properties is mainly related to the compatibility between RR particles and poly(ε-caprolactone). When the amount of PCL increases, the influence of its cohesion on the mechanical properties of modified RR is determinant. PCL acts as a continuous phase in prepared materials and the properties of the type of PCL applied are directly related to the performance of the material. Analyzing results from Table 3 one can see that with the increasing amount of PCL, the type of used polyester has a very significant influence on the mechanical performance of RR/PCL blends according to the values of tensile properties. It is evident that the properties of samples with Capa™ FB100 are mostly superior to those with addition of Capa™ 6800.

### 3.4. Dynamic Mechanical Analysis

Figure 3 shows plots of the loss tangent as a function of the temperature of the investigated samples in order to determine their glass transition temperature (T_g_). In all cases, the shape and the variations of the curves according to the composition of the blends are similar. First, all curves show only one tan δ peak. This implies that there are good compatibility and homogeneity of the materials included in the blends. In this case, the relative proximity of the T_g_ of the components of the blend also helps to the homogeneity of the curves. As happens with compatible blends, increasing the proportion of PCL in the materials produces a displacement of the T_g_, in this case, towards lower values. The glass transition temperature of the blends containing 90:10 RR:PCL is higher for more than 15 °C compared to the pure PCL.

The two types of poly(ε-caprolactone) Capa™ 6800 and Capa™ FB100 are semi-crystalline polymers and their melting point is comprised in the interval 58–60 °C. This means that the observed variations in T_g_ are compatible with the higher values of the mechanical properties that they impart to the blend with RR. The amorphous parts of the polymer can show compatibility with the RR and produce a decrease of the T_g_ of the blend towards lower values, but at the same time, the whole material can be stiff and present a high Young’s modulus. In fact, the combination of elasticity provided by the amorphous regions and the stiffness due to the crystalline areas, which results in the increased toughness observed in the mechanical tests performed (see Table 3).

Moreover, a significant decrease of tanδ peak intensity was noted related to the addition of PCL, which points to changes in viscoelastic characteristics of materials. Lower values of tan δ are characteristic of elastic materials and related to the ability of the sample to store energy, while higher values are typical of a non-elastic behavior, where dissipation of energy occurs. The non-elastic behavior is associated with specimens such as rubbers or bitumen, the components of the modified reclaimed rubber. Then, the addition of PCL produces an increase of the elastic (storage) modulus (E’) which varies from 13 to 163 MPa when the proportion of PCL increases from 10% to 50% (Table 3). The viscous (loss) modulus (E”) also increases (2.9 to 11.0 MPa) but in a proportion lower than E’, meaning that the resulting material increases its overall elastic behavior. These results corresponded with the results of the mechanical tests.

Comparing the contributions of Capa™ 6800 and Capa™ FB100, the second, given its different features and especially its partial cross-linking produces a more remarkable effect following in the same direction commented before. The obtained values of the elastic modulus (E’) vary from 17 to 191 MPa and the viscous modulus (E”) increases from 3.4 to 13.0 MPa. The differences in the mechanical properties obtained with the two types of PCL are related to these values.

Figure 4 shows the values of RR/PCL blends brittleness calculated from the results of static and dynamic mechanical analysis according to formula (1) proposed by Brostow et al. [24]. They stated that materials which show high elongation at break and high storage modulus, hence low brittleness, are able to withstand higher stress before failure. In this case, brittleness is considered as an antagonist of toughness, which, as mentioned above, measures the total amount of stress before the failure of the sample during tensile tests. Brostow et al. [24] presented the Equation (1) which relates the brittleness of the material to its toughness, based on data for a variety of materials including plastics as polycarbonate, polyethylene, and metals (5):(5)$$\tau =\frac{\left(b\pm cB\right)}{\left(1\pm aB\right)}$$
where: *τ* stands for toughness, *B* stands for brittleness, and *a*, *b,* and *c* are constants.

Brostow et al. [24] determined the values of the parameters a, b, and c, considered as “universal” constants: *a* = −111, *b* = −14,102, and *c* = −1640, with a fit parameter *R^2^* = 0.934. The previous formula can also be expressed as an exponential function, which would adopt the following form (6):(6)$$\tau =aB^{b}$$
where: *τ*, *B* has the same meaning at formula 3 and *a* and *b* are other constant parameters.

The values of *a* and *b* parameters for their data are 178.38 and −0.984, respectively. Figure 4 includes the representation of the curve, based on their data, in order to compare to the data of the composites presented in this article. The results obtained for RR/PCL blends do not correspond very well with the Brostow curve. The values of *a* and *b* parameters for a curve fitted for the RR/PCL data are 486.25 and −0.639, respectively. Comparing the two curves, for the same level of brittleness, RR/PCL blends shows higher toughness that the average of materials studied by Browtow et al. [24]. Similar observations were made in our previous work on thermoplastic elastomers highly filled with reclaimed GTR [29], where *a* and *b* parameters equaled 431.59 and −0.660, respectively. These differences can be explained by considering the type of materials studied by Brostow et al. [24], when they created their formula. Most of the materials used were homopolymers, copolymers, or metals. They did not include composite materials or blends. In this work, as well as in our previous research, we studied the properties of a combination of RR with thermoplastic materials. These structures can result in an unusual combination of brittleness and toughness caused by an overlap of the properties of its individual components. This is particularly remarkable in compositions with a high content of PCL. For example at high PCL contents, PCL becomes a continuous phase in analyzed materials, therefore during tensile tests, it is responsible for stress transfer, and due to its mechanical properties, materials modified with 50 wt.% of PCL show high tensile strength (in any case, lower than pure PCL). On the other hand, their brittleness, compared to the materials included in the Bristow’s curve is relatively high, because of the low stiffness of RR. Instead, at lower contents of PCL, the obtained results are more close to the data included in the Brostow curve [24]. Therefore, although their model is quantitatively valid for a variety of polymers and metals, it does not include the different behavior of more complex materials, which is often dependent on matrix cohesion and interfacial adhesion. Nevertheless, the shape of the presented curves is similar, so qualitatively similar dependences between toughness and brittleness can be observed.

### 3.5. FTIR Analysis

Figure 5 shows the FTIR spectra of two different PCL (Capa™ 6800 and Capa™ FB100) and bio-blends RR/PCL 90:10 and RR/PCL 50:50 in the fingerprint region between 1800 and 700 cm^−1^. Few differences appear in the infrared spectra of both PCL, mainly in the shapes of the bands. The spectra show that the intensity of the band at 1724 cm^−1^, ascribed to the stretching vibrations of carbonyl bonds, depends on the amount of PLC, where the reference samples (pure PCLs) have higher intensity. The band at 1640 cm^−1^ is related to the double bonds of the elastomeric compound (GTR) and appears in the samples with the presence of RR. Numerous absorption bands appear between 1480–1360 cm^−1^, assigned to the deformation vibrations of C–H bonds, depending on the analyzed sample. The band at 1294 cm^−1^ was associated with the stretching vibrations of C–O, and C–C bonds in the crystalline phase of poly(ε-caprolactone). The decrease of the intensity of these bands with the incorporation of RR may suggest a variation in the interactions of the crystalline structure of the PCL due to the presence of modified RR. Figure 6 shows the magnification of the spectral area from 1250 to 1000 cm^−1^, where the peak at 1240 cm^−1^ was ascribed to the asymmetric stretching of C–O–C groups and the broadband at 1190–1160 cm^−1^ is characteristic of a complex structure and is associated with the symmetric C–O–C stretching, as well as stretching vibrations of C–O, and C–C bonds in the amorphous phase of poly(ε-caprolactone). This band allows the observation of the structural differences between both types of PCL and the interactions between the PCL and RR. The magnification of this area shows a clear difference in the intensity of the peak at 1186 cm^−1^ and the displacement of the peak at 1240 cm^−1^, both attributed to C–O–C of poly(ε-caprolactone) and very sensitive to the carbonyl bands of RR or bitumen. The bands in the range of 1106–1044 cm^−1^ are attributed to the stretching of C–O bonds. The bands that appear from 730 to 963 cm^−1^ are related to the skeletal vibrations of C–C bonds in poly(ε-caprolactone) chains, where the band at 730 cm^–1^ is assigned to *cis*-configuration and the band at 963 cm^–1^ to the *trans*-configuration [30]. Moreover, the small differences between the spectra for neat poly(ε-caprolactone) and the blends are due to the physical interactions at the interphase between reclaimed rubber and poly(ε-caprolactone).

### 3.6. Thermal Stability

Figure 7 and Table 4 show the results of the thermogravimetric analysis of the samples. It can be clearly seen that the blending of RR with both types of PCL noticeably enhances the thermal stability of the RR. This is associated with the excellent thermal stability of the PCL. Higher proportions of PCL result in higher decomposition temperatures of RR/PCL blends. Even the samples with lower contents of poly(ε-caprolactone) showed thermal stability that significantly exceeded the values required by the processing conditions (120 °C). Another important difference related to the modification of the RR with PCL was the significant decrease of char residue. Neat PCL has only a 0.1–0.2% of char residue, the decomposition of PCL at 780 °C is practically complete. GTR and therefore RR contains substances as carbon black that would remain as residua in a nitrogen atmosphere and also inorganic fillers which would remain as residua even in an oxidizing atmosphere. The obtained values of residua for GTR are coherent with those published by Colom et al. [31].

Observing Figure 7, the existence of a single temperature of decomposition can be confirmed in RR/PCL blend in ratio 50:50. In the case of RR/PCL blends in ratio 90:10 two transitions appear, the first and bigger about 390 °C and the second close to 450 °C, being associated with the main components of RR (NR and SBR) respectively. It is interesting to see that the presence of RR reduces the thermal stability of both kinds of blends, mainly in samples RR/PCL 50:50.

In Figure 8, 3D FTIR spectra of gaseous products evolved during TGA analysis of the samples are shown. Samples containing 90 wt.% of reclaimed rubber noticeably present more bands and higher absorbance signals. These FTIR spectra were recorded at a temperature of ~410 °C, when the decomposition of rubber and substances included in RR has already started, hence the amount of different gaseous products generated (Figure 8 pictures b and e). PCL starts decomposing at that temperature, as shown in Figure 8 pictures a and d. The most important bands that appear at the spectra include a band around 3080 cm^−1^ ascribed to stretching vibrations of C–H bonds in unsaturated vinyl hydrocarbon compounds, which can be generated during thermal decomposition of rubber. In addition, peaks representative of out-of-plane deformation vibrations of the same bonds appeared at ~985 and 1005 cm^−1^. Another band ascribed to unsaturated vinyl hydrocarbons was observed at 1648 cm^−1^ and was related to stretching of unsaturated C=C bonds of the RR compound.

Except for these differences, qualitatively, spectra for all analyzed samples were quite similar. The most noticeable peaks were those observed around 2940, 1770, and 1167 cm^−1^, which confirm the presence of compounds with saturated hydrocarbon chains, as well as bonds between carbon and oxygen atoms. Obtained results are in correspondence with literature data [32].

### 3.7. Thermal Properties and Crystallinity

The results of differential scanning calorimetry are presented in Figure 9 and summarized in Table 5. These results show that the Capa™ 6800 sample has a higher degree of crystallinity than the PCL FB100 sample but a very similar melting temperature (T_m_). Likewise, there are also interesting differences between both samples, especially when we study the difference between the melting enthalpy and crystallization enthalpy for each sample. This value is 7.3 J/g for the Capa™ 6800 and 20.6 J/g for the Capa™ FB100. These differences are mainly due to the difference in the molecular weight and partial cross-linking of Capa™ FB100. Higher molecular weight and partial cross-linking might result in a decrease in the ΔH_m_ value because the sample requires less energy to break all secondary bonds (van der Walls and London forces) because of a reduction in the concentration of segments of a length suitable for crystallization, as described for several authors [33]. Table 5 also shows the difference in the degree of crystallinity (X_c_) between Capa™ 6800 and RR:PCL (PCL 6800) 50:50 and between Capa™ FB100 and RR/PCL (PCL FB100) 50:50 samples. In the PCL 6800 samples, there is a decrease in the X_c_ of 4.29% (sample with plain PCL vs. sample 50:50) while in the PCL FB100 samples there is an increase in the X_c_ of 5.48% (comparing the sample with plain PCL vs. sample 50:50), this difference is attributed to the effect of the particles of RR in Capa™ 6800, breaking structural regularity of the PCL and causing a decrease in the degree of crystallinity. On the contrary, in the Capa™ FB100 samples, the existence of cross-links avoids the RR particles of disordering the structure, preserving the regularity. Then, in the segments available for crystallization, which are fixed in their ends by the cross-links, the presence of particles of RR can even act as crystallinity promoters. This would explain the better mechanical, thermal and structural properties observed in partially cross-linked Capa™ FB100 compared to Capa™ 6800 samples. The thermal analysis results are in accordance with FTIR and TGA obtained data.

### 3.8. Scanning Electron Microscopy

Samples previously subjected to tensile tests were investigated by scanning electron microscopy in order to better understanding breaking mechanism. Figure 10 shows the surface perpendicular to the direction of applied stress. It was observed that, regardless of PCL grade, higher content of PCL in RR:PCL blends resulted in fibrous and developed surface, as presented in Figure 10A,C. Visible change of surface for samples with 10 wt.% of PCL (Figure 10B,D) indicating change of damage mechanism of studied samples. For samples RR:PCL(6800) 90:10 and RR:PCL(FB100) 90:10, the “strings” of PCL phase practically disappeared and the broken surface is smooth, what is characteristic behavior for elastomers. Some small gaps (marked in the images) left by the geometric particles of RR when it is removed from the PCL matrix can also be observed, especially in samples 50/50. There is a lack of hollow cavities around the RR particles, which implies compatibility between components, as found previously in the analysis of mechanical and thermal properties.

## 4. Conclusions

The properties of the blends including reclaimed GTR and two types of PCL were studied. The production of the blends required a minimum of 10% PCL in the formulation in order to be able to be processed. Adding PCL to the reclaimed rubber resulted in a material with good compatibility between the components, as assessed by the FTIR and TGA analysis. Mechanical and thermal properties combining stiffness and toughness make this blend interesting compared to other materials. It seems that the inclusion of PCL, should improve the biodegradability of studied blends, however, this assumption needs to be verified during further studies. Both types of PCL studied improved the overall properties of the reclaimed rubber. PCL FB100 with a higher molecular weight and partially cross-linked structure produced better results than PCL 6800.

## Figures and Tables

**Figure 1 polymers-12-01204-f001:**
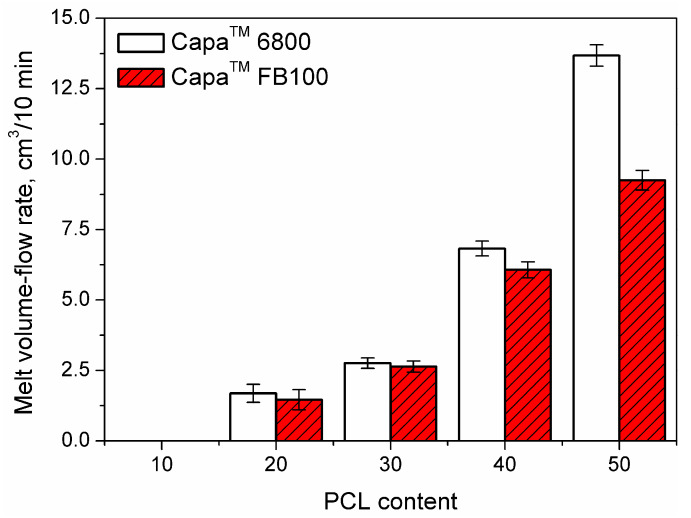
Melt volume flow rate for the studied reclaimed rubber/poly(ε-caprolactone) (RR/PCL) blends.

**Figure 2 polymers-12-01204-f002:**
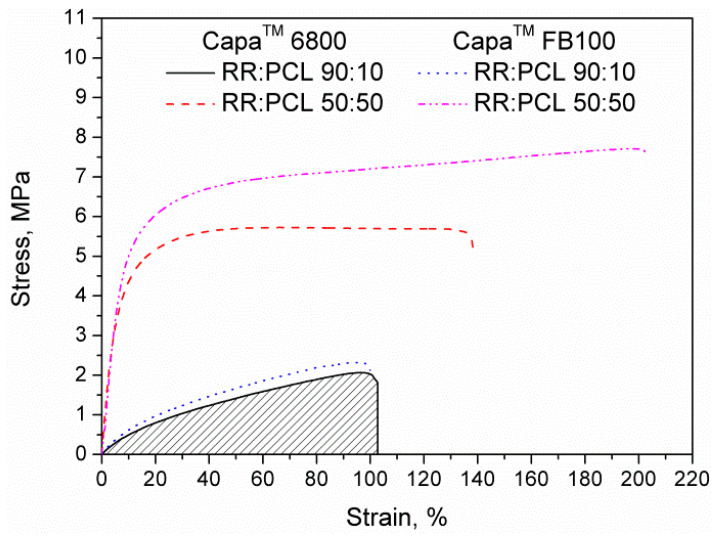
Stress–strain curves of RR/PCL blends.

**Figure 3 polymers-12-01204-f003:**
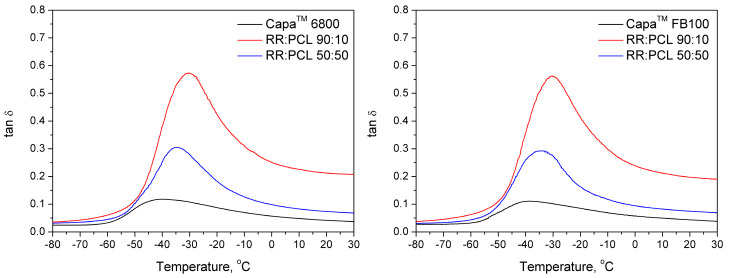
The loss tangent of the analyzed samples as a function of temperature.

**Figure 4 polymers-12-01204-f004:**
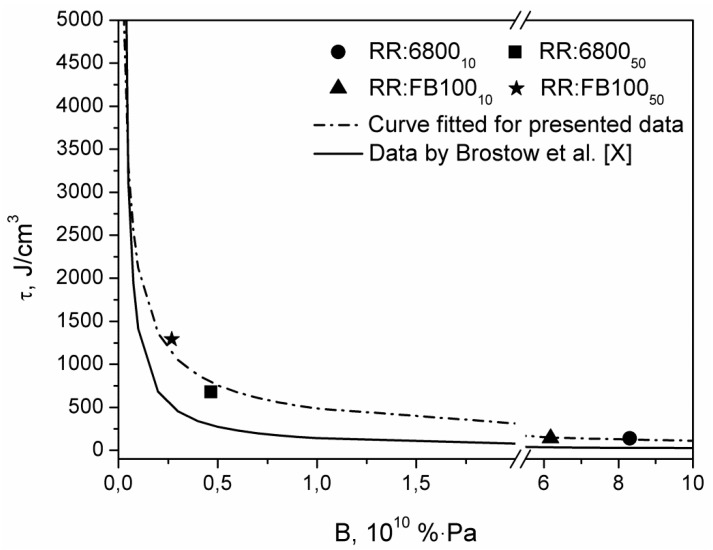
Relationship between toughness and brittleness of analyzed materials.

**Figure 5 polymers-12-01204-f005:**
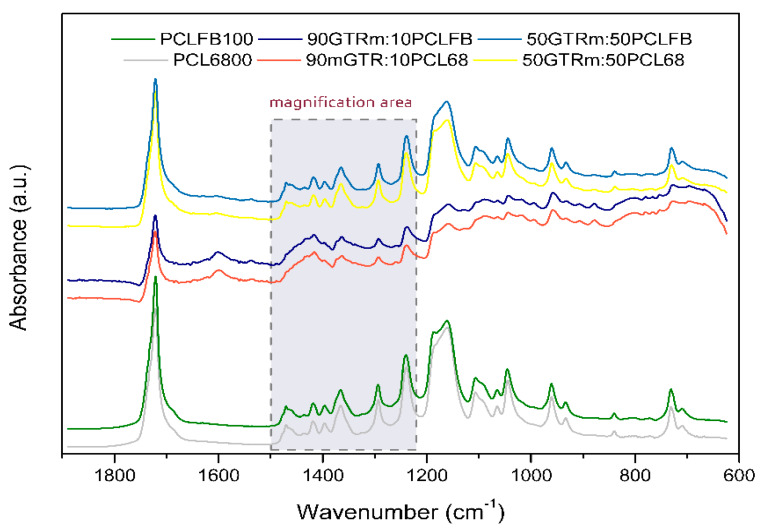
Fourier transform infrared spectroscopy (FTIR) spectra of analyzed RR/PCL blends.

**Figure 6 polymers-12-01204-f006:**
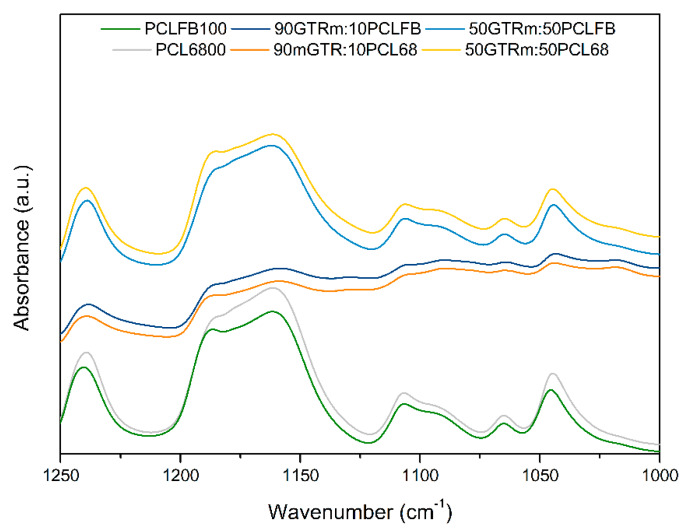
FTIR spectra of analyzed RR/PCL blends in the 1250–1000 cm^-1^ region.

**Figure 7 polymers-12-01204-f007:**
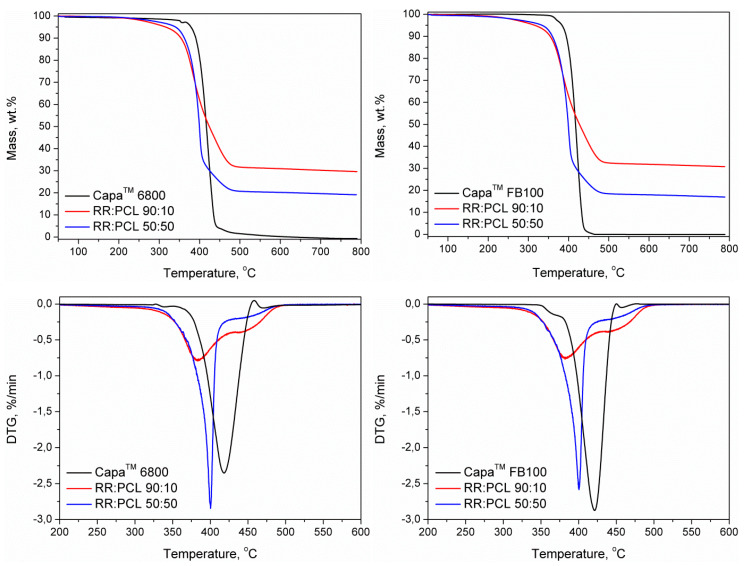
TGA and DTG curves for the composites of RR modified with (left side) Capa™ 6800 and (right side) Capa™ FB100.

**Figure 8 polymers-12-01204-f008:**
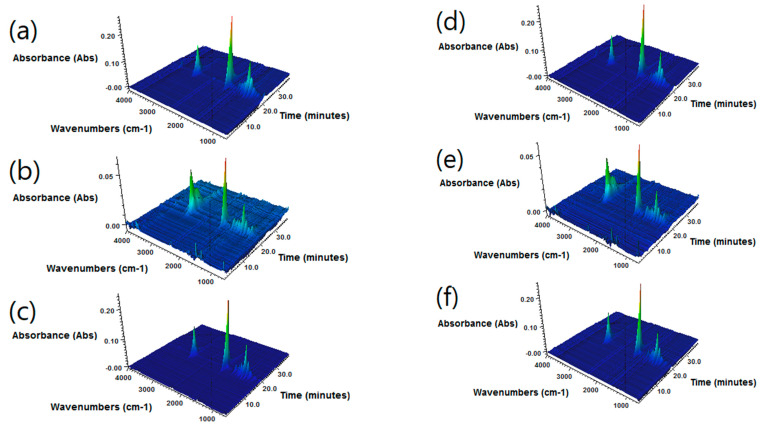
FTIR spectra of gaseous products evolved during thermogravimetric analysis (**a**) Capa™ 6800 (**b**) RR:PCL(6800) 90:10, (**c**) RR:PCL(6800) 50:50, (**d**) Capa™ FB100, (**e**) RR:PCL(FB100) 90:10, and (**f**) RR:PCL(FB100) 50:50.

**Figure 9 polymers-12-01204-f009:**
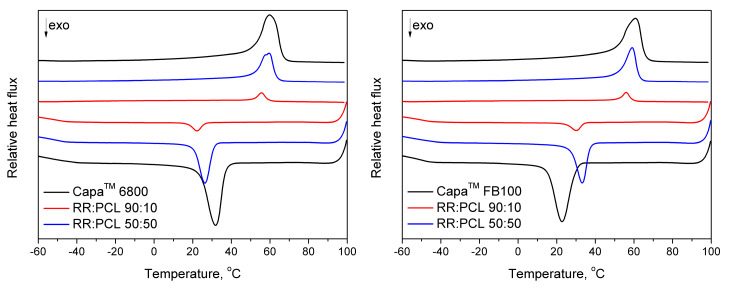
Differential scanning calorimetry (DSC) curves for the RR blended with (left side) Capa™ 6800 and (right side) Capa™ FB100.

**Figure 10 polymers-12-01204-f010:**
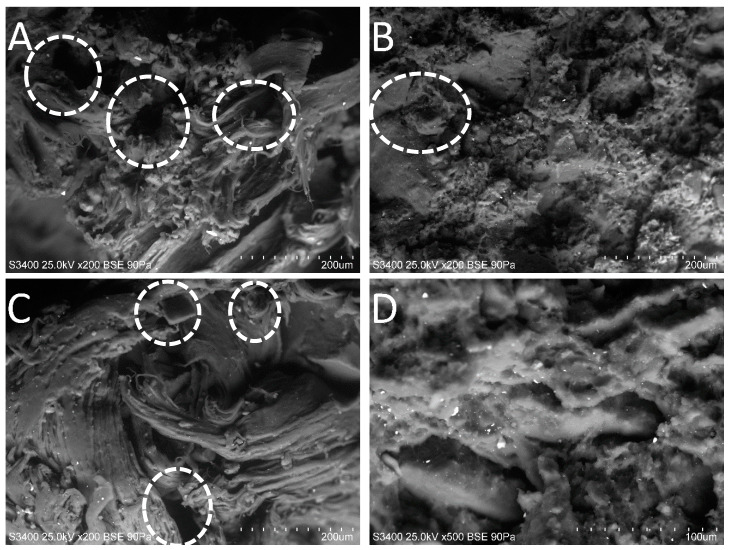
SEM images of surface after tensile test: (**A**)—RR: PCL(6800) 50:50; (**B**)—RR:PCL(6800) 90:10; (**C**)—RR:PCL(FB100) 50:50; (**D**)—RR:PCL(FB100) 90:10.

**Table 1 polymers-12-01204-t001:** Properties of the used poly(ε-caprolactone) (PCL) types.

Properties	PCL Type	
	Capa™ 6800	Capa™ FB100
Density at 25 °C, g/cm^3^	1.132 ± 0.001	1.130 ± 0.002
Density at 210 °C, g/cm^3^	0.958 ± 0.019	0.934 ± 0.041
MFR_210 °C, 10 kg_, g/10 min	30.80 ± 0.76	17.13 ± 0.43
MVR_210 °C, 10 kg_, cm^3^/10 min	32.16 ± 0.66	18.34 ± 0.71
Viscosity at 210 °C, Pa·s	11.10	19.47
Tensile strength, MPa	25.1 ± 0.8	42.5 ± 3.4
Elongation at break, %	644 ± 31	739 ± 73
Toughness, J/cm^3^	9800 ± 580	14196 ± 557
Hardness, °ShA	95.9 ± 1.3	96.9 ± 0.5
Hardness, °ShD	52.2 ± 1.6	54.9 ± 0.4

**Table 2 polymers-12-01204-t002:** Density of reclaimed rubber/poly(ε-caprolactone) blends.

PCL Type	PCL Content, wt.%	Density at 25 °C, g/cm^3^		Porosity, %	Density at 210 °C, g/cm^3^
		Experimental	Theoretical		
Capa™ 6800	10	1.144 ± 0.004	1.175	2.65	-
	20	1.142 ± 0.002	1.170	2.43	1.025 ± 0.003
	30	1.139 ± 0.001	1.166	2.28	1.007 ± 0.004
	40	1.138 ± 0.003	1.161	1.96	1.004 ± 0.002
	50	1.135 ± 0.003	1.156	1.82	1.001 ± 0.004
Capa™ FB100	10	1.147 ± 0.005	1.175	2.38	-
	20	1.143 ± 0.001	1.170	2.31	1.026 ± 0.005
	30	1.141 ± 0.001	1.165	2.06	1.011 ± 0.013
	40	1.140 ± 0.003	1.160	1.72	1.010 ± 0.003
	50	1.138 ± 0.004	1.155	1.47	1.005 ± 0.003

**Table 3 polymers-12-01204-t003:** Static and dynamic mechanical properties of RR/PCL blends.

Properties	PCL Type									
	Capa™ 6800					Capa™ FB100				
	PCL Content, wt.%									
	10	20	30	40	50	10	20	30	40	50
Tensile strength, MPa	2.1 ± 0.2	2.9 ± 0.1	3.6 ± 0.1	4.7 ± 0.1	5.8 ± 0.1	2.4 ± 0.1	3.4 ± 0.1	4.3 ± 0.2	5.7 ± 0.3	7.5 ± 0.9
Elongation at break, %	89 ± 10	95 ± 4	100 ± 5	105 ± 1	131 ± 11	92 ± 8	94 ± 8	104 ± 15	131 ± 24	195 ± 28
Permanent set, %	26.0 ± 1.1	28.9 ± 0.5	29.90.6	31.7 ± 0.7	39.3 ± 1.9	26.9 ± 0.8	28.8 ± 1.0	31.2 ± 1.2	35.9 ± 3.0	51.6 ± 1.2
Toughness, J/cm^3^	138 ± 7	213 ± 17	275 ± 22	392 ± 38	680 ± 93	141 ± 16	240 ± 29	361 ± 49	596 ± 78	1290 ± 82
Hardness, ShA	61.2 ± 0.7	73.1 ± 0.9	83.2 ± 0.5	89.3 ± 0.7	92.6 ± 0.9	63.3 ± 0.9	78.0 ± 1.0	84.2 ± 0.9	90.0 ± 0.6	92.9 ± 0.5
Hardness, ShD	14.2 ± 0.4	20.3 ± 0.4	26.1 ± 0.2	32.0 ± 0.5	36.8 ± 0.5	15.5 ± 0.6	22.1 ± 1.7	27.8 ± 0.7	33.3 ± 0.5	38.2 ± 0.6
E’ at 25 °C, MPa	13.53	-	-	-	163.98	17.59	-	-	-	191.23
E” at 25 °C, MPa	2.88	-	-	-	11.52	3.42	-	-	-	13.70
tan at 25 °C	0.21	-	-	-	0.07	0.19	-	-	-	0.07
T_g_, °C	−30.6	-	-	-	−34.6	−30.2	-	-	-	−33.3
B, 10^10^ %·Pa	8.31	-	-	-	0.47	6.18	-	-	-	0.27

**Table 4 polymers-12-01204-t004:** Results of thermogravimetric analysis of the RR/PCL blends.

Sample	Mass Loss, wt.%				Char Residue, wt.%
	2	5	10	50	
	Temperature, °C				
PCL 6800	348.5	378.7	390.7	416.8	0.1
RR: PCL 6800 90:10	255.3	316.3	353.9	424.0	29.7
RR: PCL 6800 50:50	275.9	340.7	362.1	399.4	19.2
PCL FB100	357.2	383.7	394.2	417.2	0.2
RR: PCL FB100 90:10	251.6	316.1	354.1	426.9	30.8
RR: PCL FB100 50:50	255.0	334.1	359.5	399.0	17.0

**Table 5 polymers-12-01204-t005:** Thermal properties and crystallinity of the RR/PCL blends.

Sample	ΔH_m_, J/g	T_m_, °C	W_m1/2_, °C	ΔH_c_, J/g	T_c_, °C	W_c1/2_, °C	X_c_, %	ΔT, °C
Capa™ 6800	48.70	59.8	10.2	−56.01	31.8	7.9	34.91	28.0
RR:PCL(6800) 90:10	4.24	55.6	4.3	−4.95	22.1	5.1	30.39	33.5
RR: PCL(6800) 50:50	21.36	59.5	7.0	−29.91	26.3	6.4	30.62	33.2
Capa™ FB100	40.60	60.7	8.9	−61.16	22.8	8.5	29.10	37.9
RR:PCL(FB100) 90:10	4.11	56.0	4.2	−4.88	30.1	5.5	29.46	25.9
RR:PCL(FB100) 50:50	24.12	59.1	6.2	−28.36	33.2	5.8	34.58	25.9

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
