# Peer review of "Reclaimed Rubber/Poly(ε-caprolactone) Blends: Structure, Mechanical, and Thermal Properties"

_polymers, 2020, doi:10.3390/polym12051204_

Round 1

Reviewer 1 Report

1. Author’s should add more details about RR, such as main content, filler content, etc. Because there might exist significant differences in RR from different company.

2. SEM observation of the tensile fracture surface for the samples should be provided to explain mechanical properties.

Author Response

Comments and Suggestions for Authors

  1. Author’s should add more details about RR, such as main content, filler content, etc. Because there might exist significant differences in RR from different company.

Answer: We agree with respected Reviewer. The reclaimed rubber was obtained according to our own laboratory procedure as described. Reclaimed rubber was prepared from ground tire rubber.

The main components of the ground tire rubber are natural rubber and synthetic rubber. This is confirmed by the presence of two peaks in the differential thermogravimetric analysis. Apart from those two basic components, there are the curing additives and carbon black.

The total content of polymers and organic additives is ~60-65 wt.%, while content of carbon black and inorganic additives is ~30-35 wt.% as estimated by the thermogravimetric analysis. Moreover, it should be pointed out that the precise composition of the post-consumer waste that can be included under the denomination GTR (e.g ratio between natural rubber to synthetic rubber) is variable and unknown, due to complex the composition of waste tires and the mixture of different types prior to the grinding carried out by the supplier.

The lack of detailed information about RR was also pointed by Reviewer #1. According to the suggestions of the respected reviewers we have modified this part. In the current version we clarify description of this material, as presented below:

The following two-roll mills settings were applied: friction equaled 1.08 and the gap width varied between 0.2 and 3 mm. The time of treatment was 15 minutes and the obtained reclaimed rubber (RR) showed a continuous and homogenous structure. Sol fraction of reclaimed rubber was 15 wt.%. More detailed information about the impact of bitumen on low-temperature reclaiming was published elsewhere [12, 19]. The main components of ground tire rubber are natural rubber, synthetic rubber (presence of two peaks confirmed by differential thermogravimetric analysis), curing additives and carbon black. Total content of polymers and organic additives is ~60-65 wt.%, while content of carbon black and inorganic additives is ~30-35 wt.% estimated by thermogravimetric analysis. Moreover, it should pointed out that precise composition of post-consumer waste as GTR (e.g ratio between natural rubber to synthetic rubber) due to complex composition of waste tires and their mixture prior grinding is unknown.

  1. SEM observation of the tensile fracture surface for the samples should be provided to explain mechanical properties.

Answer: According to the suggestion of the respected reviewer we have added SEM images of the tensile fracture surface. The obtained data are discussed in section 3.8, as presented below:

3.8. Scanning electron microscopy

Samples previously subjected to tensile tests were investigated by scanning electron microscopy to better understanding breaking mechanism. Figure 10 shows the surface perpendicular to the direction of applied stress. It was observed that, regardless of PCL grade, higher content of PCL in RR:PCL blends resulted in fibrous and developed surface, as presented in Figures 10A and 10C. Visible change of surface for samples with 10 wt.% of PCL (Figures 10B and 10D) indicating change of damage mechanism of studied samples. For samples RR:PCL(6800) 90:10 and RR:PCL(FB100) 90:10, the “strings” of PCL phase practically disappeared and the broken surface is smooth, what is characteristic behavior for elastomers.

see attached document

Figure 10. SEM images of surface after tensile test: A – RR: PCL(6800) 50:50; B – RR:PCL(6800) 90:10; RR:PCL(FB100) 50:50; RR:PCL(FB100) 90:10

Reviewer 2 Report

Dear Authors,

  I think that the article is well organized. However, I am not sure about the impact of adding PCL to GTR. It may be new but not sure it is worth while to doing it.

Good luck.

Author Response

Dear Authors,

I think that the article is well organized. However, I am not sure about the impact of adding PCL to GTR. It may be new but not sure it is worthwhile to doing it.

Good luck.

Answer: Thank you for your thoughtful comment. We understand the specific concerns expressed by the reviewer, because the applications of these materials are nowadays very specific, but we also believe that our presented preliminary results could be useful in further studies focused in determined uses as:

- Filtration membrane systems dedicated for gases/liquids. Since PCL improves the processing of waste rubber at lower temperature, this approach limits thermal degradation of GTR during its processing and formulation, for example, when processing the membranes by hot pressing techniques.

The possibility of using the proposed materials on these types of applications has been presented in relevant publications by other authors:

https://www.sciencedirect.com/science/article/pii/S0376738816310912

https://www.sciencedirect.com/science/arti     cle/pii/S0959652619325995

- Environmentally-friendly electrodes or low-cost flexible electronic materials. PCL improves processing of waste rubber at lower temperature, then our MFI and TGA results with higher concentration of PCL (above 20wt.%), indicate that it will be possible to use FDM technique in the formulation of such materials.

This approach is based in the studies carried out by several authors in the following publications:

https://doi.org/10.1007/s10854-019-01838-4

https://www.sciencedirect.com/science/article/pii/S0008622317305663

https://pubs.rsc.org/en/content/articlelanding/2019/lc/c9lc00417c#!divAbstract

The focus in these novel approaches and applications would allow an effective upcycling of waste tires, what constitutes currently a strongly desired direction because of the reasons that we have already exposed in the introduction of our paper. In our opinion the objective is obtaining new and highly-valuable products based on waste tires, may partially justify the relative high price of PCL as modifier.

Reviewer 3 Report

The paper presents results on the realization and testing of a reclaimed
rubber/PCL system. Some clarification and improvements are needed before it can be suitable for publication.

General comments:

PCL is mainly used in "niche" applications and has a relatively high cost. It is biodegradable but typically not produced from renewable feedstock. Could the authors better clarify the rationale of PCL addition to reclaimed rubber? Is there any foreseable use for the resulting materials?

The description of the process as "mechano-chemical" is in my opinion not well supported by data. The extent of mechanically induced devulcanization is not investigated. In the reference paper by the same authors (ref 19) some more details are provided, however, the conditions of mechanical processing are not the same (longer treatment) or not easy to compare (reference to a "narrow gap" between rolls) with the conditions used in the present paper.
The authors are invited to comment more on the effect of the mechanical treatment, also in reference to their previous investigation.

Some information on the morphology (e.g. SEM micrographs) of the prepared materials would be very useful to help discussing properties, giving details on the distribution of the PCL and rubber phase and on their adhesion. Interaction/adhesion among the phases are mentioned several times in the discussion and should be supported by morphology. Moreover, the description of phase structure is sometimes confusing. PCL is seen as covered by the rubber phase or matrix (last part of pag 5), then it is stated that PCL act as a binder for the rubbery particles (discussion of static mechanical properties), eventually becoming a continuous matrix at high PCL content. These statements should be supported by some analysis/micrographs. It is also stated that the mechanical properties increase with increasing reinforcement content (first lines of pag. 7), however if PCL is the continuous matrix, then RR is the "reinforcement", common definition for the dispersed phase in composites.

While a large number of composition was produced, some data shown refer only to the 10 and 50 wt% PCL. Data relative to (at least) one more composition should be shown in figures and tables, to allow a better understanding of the trends observed and enrich the discussion of properties. When possible, data of the reclaimed rubber should also be included (e.g. DSC and TGA curves and parameters)

Specific comments:

Introduction - the reference to zinc oxide and its interactions with PCL is not clear.

methods - cooling rate for DSC analysis in not reported

section 3.1 - The discussion of porosity with reference to different molecular structure of PCL is not clear. The presence of branching or light crosslinking in PCL has no obvious correlation to structure in such a complex mixture (see previous remark on morphological analysis) and in such a wide compositional range.

Mechanical properties - elastic modulus should be calculated, at least for the materials with high PCL content.
Table 3 caption is "Static mechanical properties..." but contains also DMA results

pag.8 last line - temperature should be "glass transition temperature"

pag.9 - the discussion of loss tangent, Tg and its relation to morphology and mechanical properties is not clear and should be better supported by either theory or experimental data. The dicussion points out a decrease of Tg in the blends, however Tg do increase with respect to PCL with increasing RR share. The Tg of neat RR is not mentioned. In the lack of DMA data, DSC could help to improve the understanding of this point as the Tg of all materials including RR can in principle be measured.

FTIR - The spectrum of RR is also needed to allow a meaningful comparative analysis between the spectra reported.

DSC - thermal data of the RR should be added to have an idea of the Tg. The analyzed range is -80 - 200 °C so the lower temperature part of the curves could be showed, to allow a better visualization of the Tg.
The discussion on the crystallization behavior of PCL is speculative, it should be supported by literature and/or by (again) morphological analysis.

Author Response

Reviewer #3

The paper presents results on the realization and testing of a reclaimed rubber/PCL system. Some clarification and improvements are needed before it can be suitable for publication.

General comments:

PCL is mainly used in "niche" applications and has a relatively high cost. It is biodegradable but typically not produced from renewable feedstock. Could the authors better clarify the rationale of PCL addition to reclaimed rubber? Is there any foreseeable use for the resulting materials?

Answer: Thank you for your thoughtful comment. We understand the specific concerns expressed by the reviewer, because the applications of these materials are nowadays very specific, but we also believe that our presented preliminary results could be useful in further studies focused in determined uses as:

- Filtration membrane systems dedicated for gases/liquids. Since PCL improves the processing of waste rubber at lower temperature, this approach limits thermal degradation of GTR during its processing and formulation, for example, when processing the membranes by hot pressing techniques.

The possibility of using the proposed materials on these types of applications has been presented in relevant publications by other authors:

https://www.sciencedirect.com/science/article/pii/S0376738816310912

https://www.sciencedirect.com/science/arti     cle/pii/S0959652619325995

- Environmentally-friendly electrodes or low-cost flexible electronic materials. PCL improves processing of waste rubber at lower temperature, then our MFI and TGA results with higher concentration of PCL (above 20wt.%), indicate that it will be possible to use FDM technique in the formulation of such materials.

This approach is based in the studies carried out by several authors in the following publications:

https://doi.org/10.1007/s10854-019-01838-4

https://www.sciencedirect.com/science/article/pii/S0008622317305663

https://pubs.rsc.org/en/content/articlelanding/2019/lc/c9lc00417c#!divAbstract

The focus in these novel approaches and applications would allow an effective upcycling of waste tires, what constitutes currently a strongly desired direction because of the reasons that we have already exposed in the introduction of our paper. In our opinion the objective of obtaining new and highly-valuable products based on waste tires, may partially justify the relative high price of PCL as modifier.

The description of the process as "mechano-chemical" is in my opinion not well supported by data. The extent of mechanically induced devulcanization is not investigated. In the reference paper by the same authors (ref 19) some more details are provided, however, the conditions of mechanical processing are not the same (longer treatment) or not easy to compare (reference to a "narrow gap" between rolls) with the conditions used in the present paper. The authors are invited to comment more on the effect of the mechanical treatment, also in reference to their previous investigation.

Answer: Thank you for this comment. The lack of detailed information about RR was also pointed by Reviewer #1. According to the suggestions of the respected reviewers we have modified this part. In the current version we clarify description of this material, as presented below:

The following two-roll mills settings were applied: friction equaled 1.08 and the gap width varied between 0.2 and 3 mm. The time of treatment was 15 minutes and the obtained reclaimed rubber (RR) showed a continuous and homogenous structure. Sol fraction of reclaimed rubber was 15 wt.%. More detailed information about the impact of bitumen on low-temperature reclaiming was published elsewhere [12, 19]. The main components of ground tire rubber are natural rubber, synthetic rubber (presence of two peaks confirmed by differential thermogravimetric analysis), curing additives and carbon black. Total content of polymers and organic additives is ~60-65 wt.%, while content of carbon black and inorganic additives is ~30-35 wt.% estimated by thermogravimetric analysis. Moreover, it should pointed out that precise composition of post-consumer waste as GTR (e.g  ratio between natural rubber to synthetic rubber) due to complex composition of waste tires and their mixture prior grinding is unknown.

Some information on the morphology (e.g. SEM micrographs) of the prepared materials would be very useful to help discussing properties, giving details on the distribution of the PCL and rubber phase and on their adhesion. Interaction/adhesion among the phases are mentioned several times in the discussion and should be supported by morphology. Moreover, the description of phase structure is sometimes confusing. PCL is seen as covered by the rubber phase or matrix (last part of pag 5), then it is stated that PCL act as a binder for the rubbery particles (discussion of static mechanical properties), eventually becoming a continuous matrix at high PCL content. These statements should be supported by some analysis/micrographs. It is also stated that the mechanical properties increase with increasing reinforcement content (first lines of pag. 7), however if PCL is the continuous matrix, then RR is the "reinforcement", common definition for the dispersed phase in composites.

Answer: We agree with the respected reviewer. We use uniform nomenclature in text, considering RR and PCL as two phases. We have also added SEM measurements, that are described in subchapter 3.8, as presented below:

Samples previously subjected to tensile tests were investigated by scanning electron microscopy in order to better understanding breaking mechanism. Figure 10 shows the surface perpendicular to the direction of applied stress. It was observed that, regardless of PCL grade, higher content of PCL in RR:PCL blends resulted in fibrous and developed surface, as presented in Figures 10A and 10C. Visible change of surface for samples with 10 wt.% of PCL (Figures 10B and 10D) indicating change of damage mechanism of studied samples. For samples RR:PCL(6800) 90:10 and RR:PCL(FB100) 90:10, the “strings” of PCL phase practically disappeared and the broken surface is smooth, what is characteristic behavior for elastomers.

see attached document

Figure 10. SEM images of surface after tensile test: A-RR: PCL(6800) 50:50; B-RR:PCL(6800) 90:10; C-RR:PCL(FB100) 50:50; D-RR:PCL(FB100) 90:10

While a large number of composition was produced, some data shown refer only to the 10 and 50 wt% PCL. Data relative to (at least) one more composition should be shown in figures and tables, to allow a better understanding of the trends observed and enrich the discussion of properties. When possible, data of the reclaimed rubber should also be included (e.g. DSC and TGA curves and parameters)

Answer: Thank you for comment. As the respected reviewer points out, a large number of compositions was produced. In order to define the trend the basic physico-mechanical measurements were determined for all compositions. After the measurements, the trend was quite clear, we observed improvement with higher content of PCL (there was no optimum in studied composition). Therefore, based on the obtained results, the more specific measurements (e.g. TGA, DSC) were performed on selected representative samples. However, some of the graphs are simplified in order to present more clearly the observed trends and to do not include an oversaturation of curves that would make tracking of the discussed points more difficult. Anyway, in the case that some of the not presented curves would have provided some extra or relevant information, we would have included them, but as stated previously, the trend is reflected with the current graphs and the reader can easily extrapolate the intermediate values.

Specific comments:

Introduction - the reference to zinc oxide and its interactions with PCL is not clear.

Answer: Thank you for comment. The reference mentioned for this point was number 18, by Varaprasad et al.

Metal-oxide polymer nanocomposite films from disposable scrap tire powder/poly-ε-caprolactone for advanced electrical energy (capacitor) applications

Varaprasad, et al. J. Clean. Prod. 2017, 161, 888-895. doi: 10.1016/j.jclepro.2017.06.002 [ref 18]

Some of the parts of the text related to the possible interaction of ZnO with PCL mentioned in our document are:

Results

FTIR:

Authors study the shifts of the characteristic peaks of the components, including PCL concluding that the observed shifts are due to interactions between the metal oxide and the polymers components of the film (including PCL) and also to the interactions in the encapsulation of the metal by the polymers.

TGA:

Authors attribute the differences in weight loss to the “interaction between the PCL, STR and the

metal-oxide nanoparticles”

methods - cooling rate for DSC analysis in not reported section

Answer: Thank you for comment. We have included these data and also rewritten this part. In the current version:

The samples of 8-9 mg weight were placed in an aluminum pan and heated from 20 °C to 100 °C under nitrogen atmosphere at a rate of 10 °C/min, subsequently material was cooled from 100 °C to -80 °C at a rate of 10 °C/min and heated to 100 °C at a rate of 10 °C/min. The results from second heating were discussed.

3.1 - The discussion of porosity with reference to different molecular structure of PCL is not clear. The presence of branching or light crosslinking in PCL has no obvious correlation to structure in such a complex mixture (see previous remark on morphological analysis) and in such a wide compositional range.

Answer: We agree with the observation of the respected reviewer. In order to provide further information about this subject we have modified the text:

The porosity of the samples has a range from 1.82 to 2.65% in the samples prepared with Capa™ 6800 and from 1.47 to 2.38% when using Capa™ FB100. Nevertheless, the influence of the type of PCL (with similar density – see Table 1) on the porosity of obtained materials was very small. The slight differences in porosity between samples obtained with the two types of PCL can be explained by complex composition of reclaimed rubber based on waste tires.

Mechanical properties - elastic modulus should be calculated, at least for the materials with high PCL content. Table 3 caption is "Static mechanical properties..." but contains also DMA results

Answer: Thank you for the comment. We have modified the caption of Table 3: Static and dynamic mechanical properties of RR/PCL blends

In our opinion, the trend presented by toughness-brittleness of the samples combined with the quite complete (in our opinion) set of data included in the paper is quite clear. We could add further information, but we think that (as expressed before) an oversaturation of data would make the paper denser and more difficult to read. We have opted for a presentation that enables a sound discussion of the results and allows to extract conclusions but at the same time preserves a relative comfortable readability.

pag.8 last line - temperature should be "glass transition temperature"

Answer: Thank you. We have included that modification in the text.

pag.9 - the discussion of loss tangent, Tg and its relation to morphology and mechanical properties is not clear and should be better supported by either theory or experimental data. The dicussion points out a decrease of Tg in the blends, however Tg do increase with respect to PCL with increasing RR share. The Tg of neat RR is not mentioned. In the lack of DMA data, DSC could help to improve the understanding of this point as the Tg of all materials including RR can in principle be measured. FTIR - The spectrum of RR is also needed to allow a meaningful comparative analysis between the spectra reported.

Answer: Thank you for this comment. Unfortunately, we cannot measure RR because it is impossible to compress and shape a sheet from the reclaimed rubber at the studied temperature (in this study we use 120 °C). According to our experience that temperature is too low to prepare a uniform material. We had to trust literature data that showed that Tg of reclaimed GTR cross-linked with sulfur is around -30 °C.

In a similar way, our FTIR equipment do not allow to perform measurements of pure RR (limitation is related to high carbon black content ~30-35%). As in many other publications, we have had to rely in the composition of blends in order to support our discussions.

DSC - thermal data of the RR should be added to have an idea of the Tg. The analyzed range is -80 - 200 °C so the lower temperature part of the curves could be showed, to allow a better visualization of the Tg. The discussion on the crystallization behavior of PCL is speculative, it should be supported by literature and/or by (again) morphological analysis.

Answer: Thank you for this comment. We would like to apologize, the analyzed range was -80°C to 100°C. Unfortunately, in studied conditions it was not possible to observe, Tg of GTR or PCL (Tg was determined by DMA method), that was related to limitation of equipment which we think is common in these type of studies.

For the sake of clarity, below we have presented the DSC curves for samples coded as RR:PCL(FB100) 90:10 (Figure A) and RR:PCL(FB100) 50:50 (Figure B)

see attached document

Round 2

Reviewer 1 Report

I think it could be accepted now.

Author Response

Thank you very much.

Reviewer 3 Report

I have to say that, although the authors addressed my comments, some answers are not completely satisfactory.

The answers on the possible applications reported for the materials produced and on the role of zinc oxide should have been used to revise the introduction, to shed more clarity on the objectives of the paper.

The request for the complete results of an additional composition was meant to corroborate the discussion of the phenomena observed, leaving to the reader the choice to go into the data in more or less detail. Likewise, I do not see how the addition of a common parameter like elastic modulus could possibly be a problem for the reader.

Finally, the SEM micrographs provided are significant for the mechanical properties but are not so clear for the determination of the morphology of PCL/RR mixtures that is, in my opinion, an important point.

Giving the lack of additional data, the authors are invited to at least improve the introduction and the discussion of morphology-related parts.

Author Response

The answers on the possible applications reported for the materials produced and on the role of zinc oxide should have been used to revise the introduction, to shed more clarity on the objectives of the paper.

Answer: Thank you for comment which will result in an effective improvement of this paper. Our main purpose is to present clearly the objectives in our publications. Following the suggestion proposed by the reviewer, we have changed a paragraph of the introduction describing better the zinc oxide-PCL interaction according to the answer provided in previous revision and also an ampliation of the possible applications of these materials including new references. Modified paragraph (marked in the text) appears as:

“Moreover, this phenomenon can explain why the addition of ZnO to PCL/GTR blends results in improved dielectric constant and dielectric loss capacities in comparison to blends with CuO or CuO0.5-ZnO nanoparticles, what was recently described by Varaprasad et al., who also describes the existence of interactions between PCL and ZnO based on FTIR and TGA results [18]. Other interesting applications of blends including GTR and PCL, are related to the possibility of improvement of the processing of the rubber waste because the addition of PCL, which would make these materials interesting for filtration membranes [19, 20] or low-cost flexible electronic materials [21, 22].  “

The request for the complete results of an additional composition was meant to corroborate the discussion of the phenomena observed, leaving to the reader the choice to go into the data in more or less detail. Likewise, I do not see how the addition of a common parameter like elastic modulus could possibly be a problem for the reader.

Answer: We understand the concern of the respected reviewer, in this case we consider that the provided data are enough to support the discussion and provide a sound basis for the understanding of the behavior of the samples. At the same time, they are in consonance with our argumentation and discussion thread. Readers that are interested in more detail can easily obtain the modulus values from the data included in the Table 3 or from the Figure 2.

Finally, the SEM micrographs provided are significant for the mechanical properties but are not so clear for the determination of the morphology of PCL/RR mixtures that is, in my opinion, an important point.

Answer: Thank you for comment. As the respected reviewer points out, in SEM microphotographs, sometimes, it is not easy to perceive the morphology details of some mixtures, mainly because of the limitations of the own technique. In this case, we can see that, in PCL / RR compounds, the small gaps that leave the geometric particles of RR when they are removed from the PCL matrix are clearly observed, especially in samples 50/50. (see the attached pictures). The lack of hollow cavities around the RR particles is also observed, which implies a certain compatibility between components. This degree of compatibility has been corroborated in the analysis of mechanical, and thermal properties.  In this sense, further explanation has been added at manuscript.  (see attached file for figure 10)

Figure 10. SEM images of surface after tensile test: A – RR: PCL(6800) 50:50; B – RR:PCL(6800) 90:10; RR:PCL(FB100) 50:50; RR:PCL(FB100) 90:10
